# Does plasmid-based beta-lactam resistance increase *E. coli* infections: Modelling addition and replacement mechanisms

**Noortje G. Godijk**[1]*, **Martin C. J. Bootsma**[1,2], **Henri C. van Werkhoven**[1], **Valentijn A. Schweitzer**[1], **Sabine C. de Greeff**[3], **Annelot F. Schoffelen**[3], **Marc J. M. Bonten**[1]

**1** Julius Center for Health Sciences and Primary Care, University Medical Center Utrecht, Utrecht University, Utrecht, the Netherlands, **2** Department of Mathematics, Faculty of Sciences, Utrecht University, Utrecht, the Netherlands, **3** Centre for Infectious Disease Control, National Institute for Public Health and the Environment (RIVM), Bilthoven, the Netherlands

* n.g.godijk@gmail.com

**Data Availability Statement:** All data files for the mathematical model are available from https://github.com/NoorGo/AdditionandReplacement.

## Abstract

Infections caused by antibiotic-resistant bacteria have become more prevalent during past decades. Yet, it is unknown whether such infections occur in addition to infections with antibiotic-susceptible bacteria, thereby increasing the incidence of infections, or whether they replace such infections, leaving the total incidence unaffected. Observational longitudinal studies cannot separate both mechanisms. Using plasmid-based beta-lactam resistant *E. coli* as example we applied mathematical modelling to investigate whether seven biological mechanisms would lead to replacement or addition of infections. We use a mathematical neutral null model of individuals colonized with susceptible and/or resistant *E. coli*, with two mechanisms implying a fitness cost, i.e., increased clearance and decreased growth of resistant strains, and five mechanisms benefitting resistance, i.e., 1) increased virulence, 2) increased transmission, 3) decreased clearance of resistant strains, 4) increased rate of horizontal plasmid transfer, and 5) increased clearance of susceptible *E. coli* due to antibiotics. Each mechanism is modelled separately to estimate addition to or replacement of antibiotic-susceptible infections. Fitness costs cause resistant strains to die out if other strain characteristics are maintained equal. Under the assumptions tested, increased virulence is the only mechanism that increases the total number of infections. Other benefits of resistance lead to replacement of susceptible infections without changing the total number of infections. As there is no biological evidence that plasmid-based beta-lactam resistance increases virulence, these findings suggest that the burden of disease is determined by attributable effects of resistance rather than by an increase in the number of infections.

## Author summary

Infections with antibiotic-resistant bacteria (ARB) are considered a major global problem. To estimate the burden of antibiotic resistance, one should know whether, at a population level, infections with ARB replace infections with non-ARB (scenario labelled

**Funding:** This research was part of the Risk and Disease burden of Antimicrobial Resistance (RaDAR) project, which was funded through the One Health European Joint Programme by the EU's Horizon-2020 Research and Innovation Programme (grant 773830) (https://ec.europa.eu/info/research-and-innovation_en).The funders had no role in study design, data collection and analysis, decision to publish, or preparation of the manuscript. M.C.J.B. and N.G.G. were funded through this grant.

**Competing interests:** The authors declare no competing interest.

replacement) or whether infections with ARB occur on top of infections with non-ARB (scenario labelled addition). With replacement, only the additional burden of infections with ARB compared to infections with a non-ARB should be attributed to antibiotic resistance. With addition, the total burden associated with infections with ARB should be ascribed to antibiotic resistance. Using *E. coli* as example, we developed a mathematical model to investigate whether seven biological characteristics of ARB, each linked to either fitness costs or benefits, cause replacement or addition. A fitness cost causes resistant bacteria to die out if other characteristics are the same as for susceptible bacteria. Only increased virulence of ARB increases the total number of infections, while other benefits of resistance lead to replacement. As there is no biological evidence that the type of resistance in *E. coli* we studied increases virulence, these findings suggest that the burden of ARB is determined by attributable effects of resistance rather than by an increase in infections.

## Introduction

It is unknown to what extent the global increase in infections caused by antibiotic-resistant bacteria (ARB) during the last decades has changed the burden of disease. ARB infections may occur on top of infections caused by non-ARB, a scenario labelled as addition [1], but they may also replace non-ARB infections. Infections caused by ARB are more difficult to treat, resulting in more adverse health outcomes [2], which increases the healthcare burden.

In the addition scenario, there is an increase in the total number of infections together with attributable harm created by those infection caused by ARB. In the replacement scenario, the increased burden of disease due to resistance results solely from the attributable harm created by resistant compared to susceptible infections, as the total number of infections remains stable. Quantifying the relative contribution of both scenarios is of critical importance for quantifying the burden of disease created by ARB. Longitudinal observational data have been used to estimate the relative importance of addition and replacement [1,3], but the validity of these approaches suffered–inevitably–from other time-dependent changes and between study-groups differences that may influence overall incidence of infections, e.g., changes in medical procedures, population age, antimicrobial stewardship, and infection prevention measures.

As shown in Fig 1 occurrence of addition or replacement of infections caused by ARB cannot be deduced from observed time trends. Fig 1A depicts a scenario with an unobserved history and observed time trends of infections. Fig 1B–1D depict three different scenarios that can explain the same observed time trend namely, addition, replacement or a combination of the two. The scenarios depicted in Fig 1B–1D have an equal number of resistant and susceptible infections and are compatible with the observed data. Changing the unobserved prevalence suggests a different interpretation on whether addition or replacement occurs. However, as the history is unobserved it cannot be decided based on time trend data whether addition or replacement has occurred. To illustrate this with real observational data, we determined the extended-spectrum beta-lactamase (ESBL) *E. coli* bacteraemia prevalence in the Netherlands between 2014 and 2019.

We, therefore, developed a mathematical model, comprising three populations, hospitalized patients, recently hospitalized patients and the general population to determine effects of various explicitly stated biological mechanisms that may change ARB carriage and subsequent infection, and thereby estimate whether these mechanisms cause replacement of or addition to susceptible infections. We start with a neutral mathematical model and in the base-case

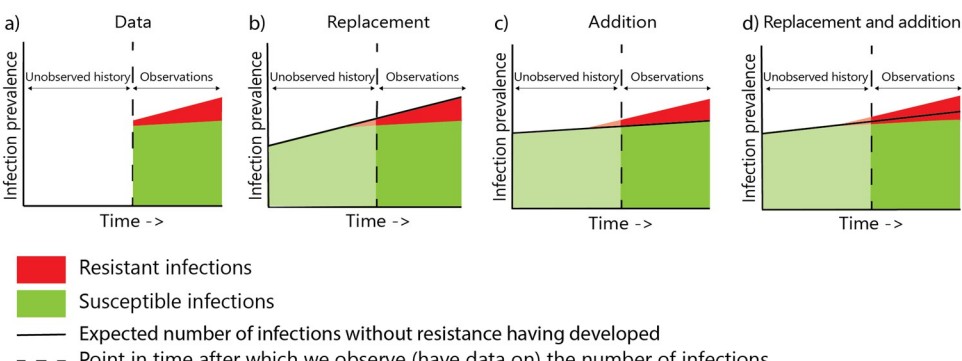

**Fig 1. Three scenarios explained by addition to and replacement of susceptible infections.** The observed prevalence of susceptible and resistant infections is depicted in panel a. In panel b, c and d, three potential scenarios for the prevalence in the unobserved period are depicted. Each scenario suggests a different interpretation regarding addition or replacement.

scenario we assume that susceptible and resistant strains can co-exist at any ratio if the two strains behave identical in all aspects. In this model, individuals either have a stable, high-density, colonization or a disturbed flora with low-density colonization. The neutral null model is further explained in the base-case scenario in the methods section and the concept of a neutral null model has also been discussed previously [4]. Next we move away from the neutral null model and modify the characteristics of the ARB strain to create benefits or costs of resistance. A fitness advantage for ARB will ultimately lead to dominance of ARB variants over non-ARB strains [5]. Multiple mechanisms, such as increased transmission or antibiotic use, can create a dynamical advantage of resistant over susceptible strains [6,7]. Similarly, resistance can be associated with a fitness cost, which fuelled the hope that natural selection would–in the absence of beneficial selective pressure—eventually lead to a reduction in ARB [8]. An example of a fitness cost could be faster clearance. Obviously, ARB (dis)advantages change in the context of antibiotic exposure. We focus on mechanisms that influence the prevalence of colonisation with ARB and non-ARB and keep host mechanisms affecting individuals' risks of acquisition of carriage or infection stable.

First, we study two mechanisms with a cost of resistance (i.e., decreased growth rates of ARB and increased clearance of ARB, with clearance defined as the rate at which individuals go from high density colonization to low density colonization), and subsequently five mechanisms that benefit ARB. The first is increased virulence, defined as an increased probability to develop infection once colonisation has been established. This increased risk of infection may result from three separate mechanisms that have the same dynamical effects in our model, such as an increased bacterial load, higher intrinsic virulence of a bacterium, and failure of antibiotic prophylaxis [6,9,10]. Moreover, increased virulence may be related to other genes harbored on the ESBL-plasmid [6]. The second is increased transmission, which leads to more ARB acquisitions and, thereby, to more subsequent infections, despite a stable infection rate [11]. The third is lower clearance, which makes ARB carriage more persistent, prolonging the risk period for infection, despite a stable infection rate [12]. The fourth is within-host plasmid transfer of resistance to susceptible strains, which increases the number of ARB strains and subsequently ARB infections. The fifth is selective antibiotic pressure, which increases probabilities of ARB acquisitions through cross-transmission due to lower density of susceptible bacteria in subjects receiving antibiotics, increasing susceptibility to acquisition with resistant

bacteria. Cross-transmission is defined here as the transfer of bacteria from a source, in this model, another individual, to a receiver which in mechanism five is the subject receiving antibiotics and subsequently having a lower density of susceptible bacteria.

Moreover, we model two scenarios of combined mechanisms. The "mixed scenario" combines increased ARB hospital transmission and increased ARB clearance in the community, thereby maximizing ARB benefits among hospitalized patients, e.g., due to high selective antibiotic pressure, and maximizing fitness costs of ARB in the absence of selective antibiotic pressure. The "double benefit scenario" combines increased ARB virulence and increased hospital transmission of ARB (through increased selective antibiotic pressure and higher contact rate by hospital staff acting as potential vectors), thereby maximizing ABR benefits in hospitalized patients.

We focus on *E. coli* with plasmid-based beta-lactam resistance, since these are widely prevalent in the population and an important cause of both community-acquired and hospital-acquired infections [13]. Moreover, ESBL and carbapenem resistant *E. coli* have been set as a critical priority for research and development by the WHO on the global list of ARB [14].

## Results

From 2014 till 2019 the ESBL *E. coli* bacteraemia prevalence in the Netherlands remained stable at around 5·4%, but the total number of *E. coli* bacteraemia increased annually with 3·4% (S1 Table and Fig 2). Using the scenarios depicted in Fig 1, the observed time trend could reflect both addition to and replacement of susceptible *E. coli* bacteraemia (Fig 2).

In the neutral model, of 100,000 subjects, 177 were hospitalised, 694 were former patients and 99,129 were in the community. Numbers of plasmid-based beta-lactam resistant and susceptible *E. coli* infections were 122 and 2,320 per 100,000 subjects annually. Infections caused by ARB occurred predominantly in former ($n = 67$) and hospitalised patients ($n = 34$), and 21 occurred in the community. Infections by non-ARB also occurred predominantly in former($n = 1,275$) and hospitalised patients ($n = 651$), and less frequently in the community ($n = 395$). The infection incidence caused by ARB and non-ARB together is 8% per year in the hospital, 16% in former patients and 0.5% in the community, yielding an infection incidence in the population (former patients and the community) of 2% per year.

The effects of changing rates related to costs and benefits of ARB in 10 years are depicted in Fig 3. Naturally, introducing costs of resistance without benefits of resistance reduces the carriage of resistant strains and eventually leads to elimination of resistant strains (S4 Fig and S7 Table). Whereas introducing a benefit without costs of resistance leads to replacement or addition.

Increased ARB virulence is associated with a linear increase in the incidence of infections caused by ARB: a 10% increased virulence leads to a 10% increase of infections caused by ARB

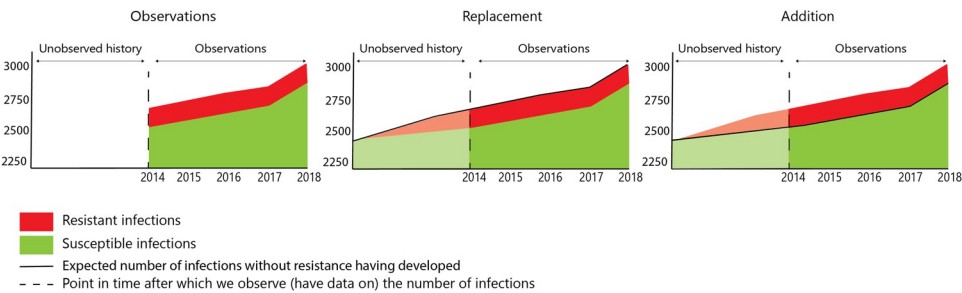

**Fig 2. Number of susceptible E. coli and ESBL *E. coli* bacteraemia in 24 Dutch hospitals from 2014 to 2018.**

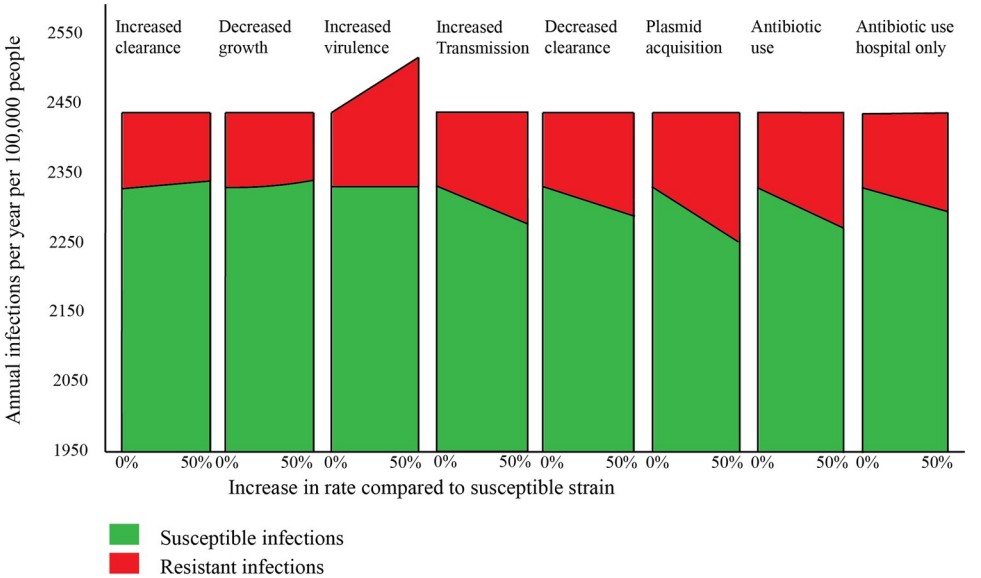

**Fig 3. Annual susceptible and resistant infections per 100,000 people per mechanism in 10 years of time.**

with an unchanged incidence of infections caused by non-ARB. Thus, increased virulence leads to an equivalent addition to the total number of infections, as can be seen in Fig 3. The transmission rates, e.g. from S to R or R to RR, are not affected by increased virulence and the subsequent increase in the incidence of ARB infections. Because the transmission dynamics are not altered by increased virulence, resistant and susceptible strains coexist in our model, which precludes eradication of susceptible strains.

The other four mechanisms associated with a dynamical benefit of ARB, all do lead to more infections caused by ARB, but only through replacing infections caused by non-ARB. In fifty years, a 10% increase in ARB transmission rates leads to a 13% increase in infections caused by ARB and a 50% decreased clearance of ARB leads to a 72% increase in infections caused by ARB which increases the prevalence of carriage with ARB from 5% to 8·6%. The total number of infections remains stable. Introduction of plasmid acquisition also causes a linear replacement of infections caused by non-ARB. In 10 years, the number of infections caused by ARB increases with 67%.

Administering a new course of antibiotics that clears 50% of non-ARB *E. coli* results in replacement of non-ARB. Restricting the use of a course of antibiotics in hospitalized patients only, reduces replacement of infections caused by non-ARB (Fig 3).

Contrary to all other mechanisms, effects of increased virulence are not time dependent, as shown in the equal trends in S2 and S3 Figs. This implies that an increase in virulence leads immediately to additional infections and this increase then remains stable. Other mechanisms benefitting ARB will lead to complete replacement of non-ARB, whereas increased virulence will lead to more infections without extinction of infections caused by non-ARB (S3 Fig and S6 Table). Costs of resistance will lead to replacement of ARB.

Thereafter, we investigated two scenarios with combined mechanisms. The mixed scenario provides maximum benefits for ARB among hospitalized patients, but also maximum fitness costs of resistance in the absence of selective antibiotic pressure. In this scenario the incidence of infections caused by ARB declines with 18% (after 50 years), whereas the incidence of infections caused by non-ARB increases (with 1%), leaving the total number of infections unaffected. In the double benefit scenario (with double benefits for ARB due to increased virulence and more transmission in hospitals) the incidence of infections caused by ARB would increase

with a factor 5·75 (after 50 years), at the cost of infections caused by non-ARB (-25%), and with a 5% increase in the total number of infections (S3 Table).

Finally, we investigated the effect of decreasing antibiotic use by 50% in all three populations on the number of infections under the conditions that each 1) mechanisms was changed by 50% and 2) by 75%. The largest decrease in resistant infections was observed for increased transmission, decreased growth, and increased clearance. Under all conditions, decreased antibiotics use lowered the number of resistant infections. The results can be found in S9 Table and S5 and S6 Figs.

## Discussion

The modelling of different biological mechanisms affecting *E. coli* with and without plasmid-based beta-lactam resistance revealed that increased virulence is the only mechanism that would lead to a higher burden of infections due to emerging antibiotic resistance. Such an increase in the incidence of infections would occur instantaneously and linearly, without affecting the number of infections caused by non-ARB. All other biological mechanisms investigated, such as increased transmission, decreased clearance, plasmid transfer, and selection through antibiotic use resulted in scenarios in which infections caused by ARB would increase at the cost of infections caused by non-ARB, with a stable overall incidence of infections. This can be explained by increased virulence being the only mechanism which increases the infection rate. The other mechanisms influence parameters affecting colonisation, and subsequently affect the ratio at which resistant and susceptible infections occur. A difference in the ratio of colonisation with susceptible and resistant strains will lead to an equal number of total infections if the infection rate for resistant and susceptible strains is assumed to be equal, because we start with a neutral null model. Moreover, if resistance has both costs and benefits, emergence may lead to a stable co-existence of susceptible and resistant strains, for instance if resistance is beneficial in one niche, e.g., the hospital, but disadvantageous in the community, as reflected in the mixed scenario.

The starting point of this work was the recognition of difficulties in interpreting observational longitudinal data on antibiotic resistance prevalence. In the Netherlands the incidence of *E. coli* bacteraemia increased over time towards a stable prevalence of ESBL-producing *E. coli* bacteraemia (Fig 2). Yet, where the prevalence of ESBL-producing strains among 173 *E. coli* bacteraemia isolates was zero around 2000 in the Netherlands [15], it was around 5% in 2014. With a stable population size and unchanged hospital policies related to infection diagnosis the 5% increase in resistance in-between both time points could have resulted from both addition and replacement mechanisms.

The biological plausibility of some of the biological mechanisms that we studied is not obvious. For instance, there is no biological prove that resistance in itself increases virulence [8]. If ARB would not have increased virulence, the possibility of the addition scenario would be excluded as we found that only increased virulence results in additional infections. Furthermore, the dogma that antibiotic resistance, such as ESBL-production in *E. coli*, is associated with fitness costs has been questioned [6,8,16]. Indeed, if resistance comes without fitness costs through, for instance, increased clearance or decreased growth, ESBL-producing *E. coli* will not to be completely replaced by susceptible *E. coli* and will persist, even in the absence of antibiotic selective pressure.

Two previous studies attempted to disentangle whether a longitudinal increase in the incidence density of infections caused by ARB resulted from replacement or addition. Firstly, de Kraker et al. [17] investigated bacteraemia trends of five pathogens; *Staphylococcus aureus*, *E. coli*, *Streptococcus pneumoniae*, *Enterococcus faecalis* and *Enterococcus faecium*, using the European Antimicrobial Resistance Surveillance System (EARSS) database from 2002 to 2008. They report an increasing incidence in bacteraemia and suggested that this was mainly driven

by resistant strains, implying that resistant clones add to rather than replace infections caused by susceptible bacteria. Secondly, Ammerlaan et al. [1] investigated temporal trends in annual incidence densities (events per 100,000 patient-days) of nosocomial bloodstream infections caused by methicillin-resistant Staphylococcus aureus (MRSA), ARB other than MRSA and non-ARB in 14 hospitals between 1998 and 2007. Seven hospitals had high incidence of MRSA infection in 1998 and no specific program to control the spread of MRSA. The other seven hospitals had low incidence of nosocomial infections with MRSA and infection control programs to maintain low incidence levels of MRSA over this period. During the 10-year period the increase in the incidence density of non-ARB infections was similar in both hospital groups. Yet, the incidence of infections caused by ARB increased from 3.0 to 4.7 per 100,000 patient-days between 1998 and 2007 in hospitals that effectively controlled ARB infections, and from 4.6 to 29.1 per 100,000 patient-days between 1998 and 2007 in hospitals with pre-existing higher MRSA and ARB infection rates. From this the authors concluded that ARB infections were additive to non-ARB infections. The observed increase in incidence density of ARB was 10-fold higher than in non-ARB. Based on the findings in the current study, for such an increase to be fully explained by addition would require a similar (i.e., linear) increase in bacterial virulence (S5 Text and S8 Table). It, therefore, seems more plausible that also in these hospitals replacement had occurred rather than addition.

A limitation of our study is that colonisation density is not modelled quantitatively, precluding modelling of within-host competition between strains. Yet, dynamics of within-host competition have not been accurately determined and it is, therefore, unknown, whether addition of this component improves modelling of the population level colonisation prevalence and infection incidence. Further, we limit acquisition routes of plasmid-based beta-lactam resistant *E. coli* and *E. coli* to humans, neglecting other reservoirs, such as livestock. Recent findings reported by Mughini-Gras et al. [18] imply that–at least in the Netherlands—the human reservoir is largely responsible for transmission among humans, with relatively little contribution of the animal reservoir.

In our study we explicitly investigate the effects of multiple mechanisms on the incidence of infections caused by ARB using a single model, which can also be used for other pathogens and in which other mechanisms can be included. In the current analyses the assumption that all individuals carry *E. coli* is vital. Investigating dynamics of pathogens for which this assumption does not hold, such as *Staphylococcus aureus*, would require an additional compartment for uncolonised subjects. Moreover, as we assume that everyone is colonised with *E. coli*, a mere increase in prevalence of the resistant strain, assuming that characteristics leading to infection are equal to the susceptible strain, does increase the total number of infections. An increase in prevalence of the resistant strain may cause an increase in the total number of infections if a bacterium is assumed not the be carried by everyone, as those uncolonised are not at risk for infection and become at risk once colonised.

Increased virulence of ARB, compared to non-ARB, is the only mechanism studied that would increase the total number of infections due to emergence of ARB. Other mechanisms, such as increased transmission, decreased clearance, plasmid transfer, and antibiotic use result in replacement of non-ARB infections by ARB infections, with a stable incidence of the total number of infections.

## Methods

### Time trend data

As an illustration of the use of time trend data to estimate addition and replacement, we estimated changes in the prevalence of ESBL in community-acquired *E. coli* bacteraemia in the

Netherlands from 2014 to 2018. We used data from the national surveillance system of antimicrobial resistance (ISIS-AR) based on routinely collected data from medical microbiological laboratories [19]. We selected blood-isolates containing *E. coli* (one isolate per patient per year) with a sample moment maximum two days after hospital admission, or sampled at the emergency department or outpatient clinic, to define community-acquired bacteraemia. ESBL *E. coli* were determined based on ESBL-confirmatory tests or resistance to cefotaxime/ceftriaxone and/or ceftazidime, according to local practice. For the analysis, 24 hospitals with complete data available for the total study period were used.

## Theoretical framework

To study the effects of seven mechanisms on the number of *E. coli* and plasmid-based beta-lactam resistant *E.coli*, we use a continuous time deterministic model, similar to Cooper et al. [20], consisting of a hospital population, a population of recently discharged patients, and the community (Fig 4A). S2 Table contains the parameters and definitions per mechanisms and S4 Table provides the parameters of transition pathways between the compartments per population. The total population size is constant and we ignore death and birth rates as these are substantially smaller than transmission and decolonisation rates in most parts of the population. Hospitalized patients are discharged to the former patient population at a constant rate. Subjects are hospitalized at constant rates, with a higher rate in the former patient population. Former patients transit to the community at constant rates, after which their hospital admission rate becomes lower.

We assume homogeneous mixing in the hospital, allowing transmission between hospitalized patients at a rate four times higher than in the community due to frequent contacts with healthcare workers and medical devices that may act as vectors for transmission. Among former patients the transmission rate is assumed to be higher compared to the community, also because of interactions with healthcare workers [21]. Further, transmission occurs between and within the former patient and community, which is one homogeneously mixed

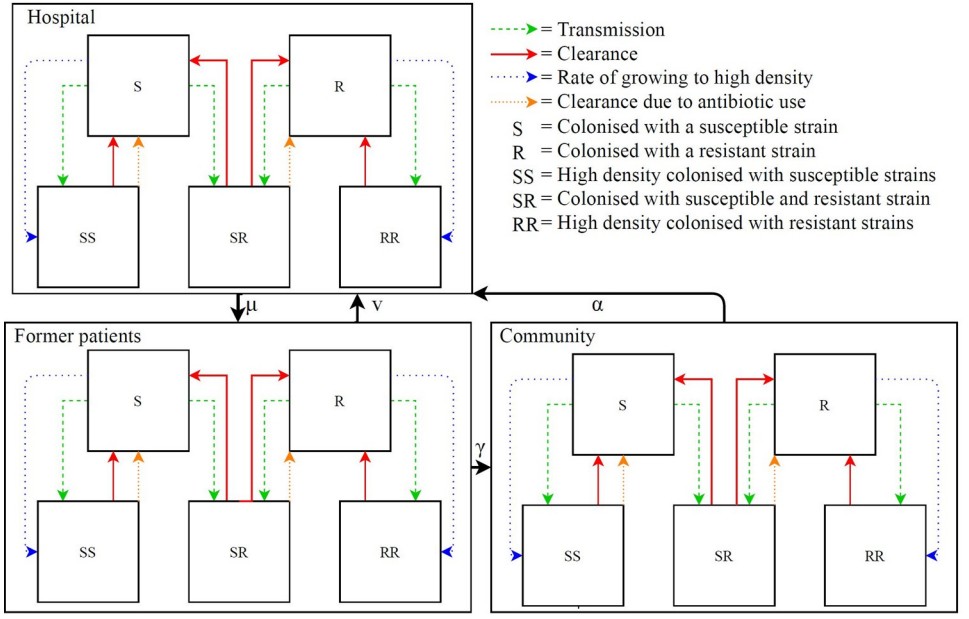

**Fig 4. Compartmental model of ESBL *E. coli* and *E. coli* colonisation states.**

population, but we assume no transmission between the community and the hospital. We use frequency-dependent transmission based on the total fraction of colonised individuals. Mathematically, it means that the rate at which an individual acquires an *E. coli* strain is proportional to the fraction of individuals colonised with that strain. Moreover, the three populations have equal ARB decolonisation rates and transition rates between populations are independent of colonisation status. S4 Table presents the model parameters.

## Base-case scenario

We developed a neutral null base-case model for resistant and susceptible bacteria which are indistinguishable except for their antibiotic susceptibility. In a neutral null model neither strain can have a selective advantage and the prevalence ratio of the two strains should remain constant over time [4]. In our base-case, we assume no antibiotic use and, therefore, the base-case model lacks an intrinsic mechanism to promote stable coexistence between strains. When we consider other scenarios, we explicitly state the selective (dis)advantages of a strain [4]. We choose a starting ARB colonisation prevalence of 5% in all populations [22].

*E. coli* is a commensal bacterial species but pathogenic variants also exist. Due to, amongst other ways, point mutations and plasmid transfers *E. coli* strains can vary in their characteristics. In our model, we assume only two strains, a susceptible and a resistant strain. Carriage with one or two of the strains is possible, but we assume everyone always carries at least one *E. coli* strain. Hence there is no compartment of uncolonized subjects and individuals can be colonized with resistant strains (R-compartment), susceptible strains (S-compartment) or both (SR-compartment). Individuals in the S-compartment may acquire ARB and then move to the SR-compartment. Similarly, individuals in the R-compartment move to the SR-compartment after acquisition of susceptible strains. To maintain neutrality in absence of selective advantages, we added SS and RR compartments [4] which we interpret as states with a high density of colonization. Without these compartments, introduction of ARB into a population in which everyone could acquire ARB would lead to an equilibrium prevalence of 50%, even in absence of selective advantages [23]. Therefore, subjects colonised with susceptible (S) or resistant strains Ⓡ remain susceptible to other strains as long as they are "low density colonised". Once progressed to high density colonised (labelled SS, RR or SR) they are non-susceptible to acquisition. We thus interpret the high-density RR, SR and SS-compartments as stable intestinal flora, preventing colonisation with other bacteria. We stress that high density colonisation is not the same as having an infection.

Infection rates are equal in high- and low-density compartments and proportional to the number of subjects colonised with resistant and susceptible strains. The infection rate is defined as the number of infections per day per person. Infections in the SR-compartment occur at an infection rate which is the average of the infection rate of the S- and R-compartment. The proportion of the infections in the SR-compartment, which are caused by susceptible bacteria and by resistant bacteria is proportional to the infection rate in the S- and the R-compartment, as can be seen in formula 16 and 17 in S1 Text. Differential equations specifying these models are given in S1 Text. *E. coli* colonisation states are depicted in Fig 4B. The parameters used in the model can be found in S4 Table.

## ARB benefits and costs

To investigate the dynamical effects of benefits and costs of resistance, we distinguish seven mechanisms, and determine the effects by gradually increasing or decreasing the applicable rates with 10% to 100%, compared to the rate in the neutral model. Mechanisms benefitting resistant strains increase the ARB colonisation prevalence. As we assume a maximum colonisation capacity, meaning that new strains cannot successfully colonize individuals in state RR,

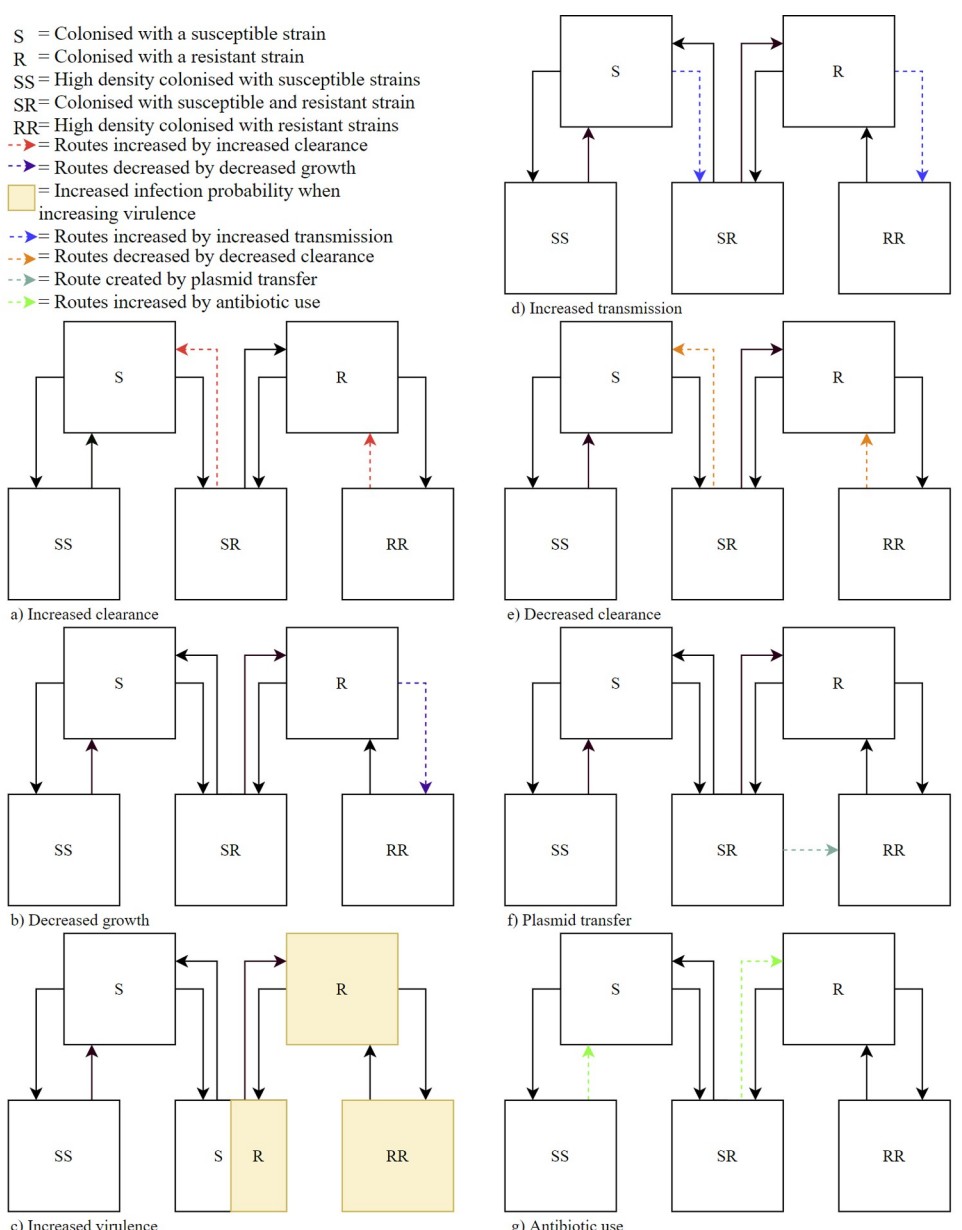

**Fig 5. The biological mechanisms shown in the compartmental model.**

SR or SS, an increased ARB prevalence renders less people susceptible for acquiring new colonisation with susceptible strains. Details of these biological mechanisms are described in S2 Table. The influence of each mechanism on the paths in the compartmental model is shown visually in Fig 5.

The mechanism of plasmid transfer was restricted to within-host and within-species transfer, thereby increasing the transition from SR to RR carriage status. Additionally, a scenario with plasmid transfer from other pathogens to *E. coli* was analysed, see S4 Text and S1 Fig.

Antibiotic use occurs in all populations—independent of infection. In the Dutch community antibiotic courses are prescribed at a rate of 0.374 per year per person [24], and we assumed that 50% of persons receiving a course of antibiotics clear susceptible *E. coli*.

Conceptually, this moves a subject from SS and SR to S and R compartments, respectively, after which the likelihood of moving to SS, RR and SR increases (Fig 4B). For hospitalized patients we assumed that 33.8% received antibiotics on any given day, based on published data [24,25]. We did not consider repeated prescriptions in the community or during hospitalization, as the intestinal flora will already be influenced by the first course of antibiotics. For changes in antibiotic use we also investigate a scenario with daily antibiotic use of 33.8% of all Dutch hospitalized subjects [25] and without antibiotic use in former patients and the community. Furthermore, we modelled two scenarios of combined mechanisms. The "mixed scenario" combines 20% increased ARB hospital transmission with 20% increased ARB clearance in the community, thereby maximizing the benefits of ARB among hospitalized patients, e.g., due to high selective antibiotic pressure, and maximizing fitness costs of ARB in the absence of selective antibiotic pressure. The "double benefit scenario" combines 20% increased ARB virulence (through increased selective antibiotic pressure) and 20% increased hospital transmission of ARB (through both increased selective antibiotic pressure and higher contact rate by hospital staff acting as potential vectors), thereby maximizing the benefits of ARB in hospitalized patients. Scenarios and mechanisms are studied during time periods of 10 years, 50 years and infinite time. The outcome is the annual number of infections after the subsequent time period (S5, S6, and S7 Tables and S2, S3, and S4 Figs).

## Supporting information

**S1 Text. Differential equations of the mathematical model.**
(DOCX)

**S2 Text. Infection rate calculations.**
(DOCX)

**S3 Text. Calculations admission and readmission rates.**
(DOCX)

**S4 Text. External plasmid transfer.**
(DOCX)

**S5 Text. Calculations of increased virulence.**
(DOCX)

**S1 Fig. Compartmental model of *E. coli* with plasmid transfer from other sources.**
(DOCX)

**S2 Fig. Annual ESBL *E. coli* infections per 100,000 people per mechanism in 10 years of time.**
(DOCX)

**S3 Fig. Annual ESBL *E. coli* infections per 100,000 people per mechanism in in 50 years of time.**
(DOCX)

**S4 Fig. Annual ESBL *E. coli* infections per 100,000 people per mechanism when letting time run to infinity.**
(DOCX)

**S5 Fig. Annual ESBL *E. coli* infections per 100,000 people with 50% change in mechanisms and decreased antibiotics use.**
(DOCX)

**S6 Fig. Annual ESBL *E. coli* infections per 100,000 people with 75% change in mechanisms and decreased antibiotics use.**
(DOCX)

**S1 Table. Annual prevalence of ESBL in *E. coli* bacteraemia in 24 Dutch hospitals.**
(DOCX)

**S2 Table. Scenarios and mechanisms studied in this paper.**
(DOCX)

**S3 Table. In 50 years, the number of resistant and susceptible infections per 100,000 inhabitants under different scenarios.**
(DOCX)

**S4 Table. Parameters used in the model.**
(DOCX)

**S5 Table. In 10 years, the number of infections per 100,000 people per year for each mechanism.**
(DOCX)

**S6 Table. In 50 years, the number of infections per 100,000 people per year for each mechanism.**
(DOCX)

**S7 Table. Letting time run to infinity, the number of infections per 100,000 people per year for each mechanism.**
(DOCX)

**S8 Table. Observed growth reported in Ammerlaan et al. [4] and expected growth.**
(DOCX)

**S9 Table. In 10 years, the number of infections for 50% and 75% change per mechanism when using antibiotics and when decrease overall antibiotic use with 50%.**
(DOCX)

## Author Contributions

**Conceptualization:** Noortje G. Godijk, Martin C. J. Bootsma, Marc J. M. Bonten.

**Data curation:** Noortje G. Godijk.

**Formal analysis:** Noortje G. Godijk, Sabine C. de Greeff, Annelot F. Schoffelen.

**Funding acquisition:** Martin C. J. Bootsma, Marc J. M. Bonten.

**Investigation:** Noortje G. Godijk.

**Methodology:** Noortje G. Godijk, Martin C. J. Bootsma.

**Project administration:** Noortje G. Godijk.

**Supervision:** Martin C. J. Bootsma, Marc J. M. Bonten.

**Visualization:** Noortje G. Godijk.

**Writing – original draft:** Noortje G. Godijk.

**Writing – review & editing:** Martin C. J. Bootsma, Henri C. van Werkhoven, Valentijn A. Schweitzer, Sabine C. de Greeff, Annelot F. Schoffelen, Marc J. M. Bonten.

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
