## [Decision Letter · Decision Letter 0]

12 Aug 2021

Dear Mrs. Godijk,

Thank you very much for submitting your manuscript "Does plasmid-based beta-lactam resistance increase E. coli infections: Modelling addition and replacement mechanisms" for consideration at PLOS Computational Biology.

As with all papers reviewed by the journal, your manuscript was reviewed by members of the editorial board and by several independent reviewers. In light of the reviews (below this email), we would like to invite the resubmission of a significantly-revised version that takes into account the reviewers' comments.

We cannot make any decision about publication until we have seen the revised manuscript and your response to the reviewers' comments. Your revised manuscript is also likely to be sent to reviewers for further evaluation.

Sincerely,

Roger Dimitri Kouyos

Associate Editor

PLOS Computational Biology

Virginia Pitzer

Deputy Editor-in-Chief

PLOS Computational Biology

Reviewer's Responses to Questions

**Comments to the Authors:**

Reviewer #1: The authors are investigating with the help of mathematical modelling whether the presence of an antibiotic resistant strain causes an increase in the total number of infections within a population in comparison to a population with only a sensitive strain present or whether the total number of infections is stable and sensitive infections are just replaced by resistant infections. For this investigation different characteristics of resistant strains are considered and their effect. As an exemplary pathogen E.coli is examined with and without plasmid-mediated antibiotic resistance and the model is parameterized according to Dutch health care data.

The work covers interesting aspects and shows relevance for a topic of high interest: estimating the burden of antibiotic resistant bacteria. However, I had difficulties with the interpretation of the model and comprehending certain model choices. I think the paper would benefit in some parts with a more detailed explanation and justification of model choices. Generally, it is not an easy task to explain structural neutral epidemiological models, in the following I attach comments, which may be of aid to the authors.

1) As the colonization of the intestinal flora is considered, one patch, I would find the model structure described in equation (14) in the Lipsitch paper easier to interpret biologically as it is hard to see a biological correspondence of the single double colonized compartments if the intestine is not considered as multiple patches (co-infection through infection of multiple patches of one host). Furthermore, in a structure like (14) there would be an interpretable transmission route from the co-colonised compartment to the single colonised compartment.

I am a bit confused by the terminology of “low density colonised” and “high density colonised”, for me this suggests for example that the SR (or SS) compartment has a higher total population of bacteria than the S / R compartment, which should not be the case in a neutral model considering one patch that can be colonized by indistinguishable strains. In L. 285 a maximum colonisation capacity is assumed, which seems to be higher for the (SS, RR, SR) compartments than for S and R. In the simplest case the intestine can be interpreted as one environment, the carrying capacity for E.coli should therefore be the same for the double colonised compartments and the singly colonised compartments, except if we assume that the intestine can be seen as two patches. If two patches are assumed in this paper, then further explanation would be helpful.

2) If we have two indistinguishable strains (neutral model) shouldn’t the clearance rate be the same for resistant and non-resistant bacteria? (Referring to table S4, where it is described how rates should look like in order to get a neutral model)

3) It took me quite a while to understand what is actually assumed by clearance, as not the clearance of infection or colonization is referred to as clearance, but the transition of a double colonised compartments to single colonised compartments.

Minor comments:

1) L.73 the short form ESBL E. coli is used without introducing it before

2) It is the first time where the neutral mathematical model is mentioned (besides the abstract) I think here a reference to a part of the text where a neutral model is explained in more detail would be helpful, as I expect some readers not to be familiar with the concept

3) L. 82/83 would be nice to name an example for the multiple mechanisms

4) L. 98 definition of cross-transmission?

5) L. 92 reference (6) regarding beta-lactamase-plasmids increasing virulence is cited as a reference on how virulence can be increased, but in the abstract there it is stated that there is no biological evidence that plasmid-based beta-lactam resistance increases virulence. From what the authors describe in the text I understood that the resistance genes itself do not cause an increased virulence. If plasmid-based resistance is considered, can the resistance gene be looked at as a separate unit? Should not the other additional genes which are on the plasmid that cause higher virulence be associated to the resistance?

6) When introducing the ODES, a reference to table S4 would be helpful to see the definitions of the parameters and the nomenclature/description of compartments

7) Supplement end of the ODE: yr_rro typo?

8) Maybe I have overseen it, but a mathematical description of the number of people that get infected (infection + cure of infection) and to see where the rate Omega was used would be nice for understanding

Reviewer #2: Thank you for the opportunity to read this paper on understanding whether resistant infections in high carriage prevalent commensal bacteria such as E coli emerge in addition to or instead of susceptible infections. The paper offers one of the first (if not the first?) modelling approach to a really difficult question, and I am interested in what it finds.

For me, the modelling presentation could be tightened up that would improve readability and interpretation.

I also have a few queries about the approach and the findings on which I would appreciate the author responses.

1. The testing of the hypotheses is highly conditional on the set up of the model. However, I found the communication of the different models to be very confusing, with a lot of text description and poor connection with the equations given in the Appendix that were not well formatted or particularly well defined. As the model structures are so important to the interpretation of the conclusion of the paper, I would strongly advise the authors to do all their hard work justice by putting together a main text multi-panel compartment diagram that intuitively and quantitatively describes their models (see under Specific comments below for some suggestions).

2. The research question concerns the incidence of infections, rather than carriage prevalence. It may be a simple point, but it took me a while to understand the set up of the model because everyone in the population is assumed to be colonised, either singly or dually. I would like the authors to be a little more explanatory in their motivation within the main text. That is, the typical resistance data we have characterise the resistance frequencies of infections rather than that of carriage. To observe increased numbers of resistant infections we could imagine them either replacing sensitive infections or adding to the total infections (as per Figs 2-3). Alternatively, we could imagine an increase in the number of resistant infections because of an increase in the carriage prevalence of resistant strains (again, either instead of or in addition to sensitive strains). Is the second option not supported by data? The authors will know this better than me, but I think that this alternative should be spelled out and reasons given for why this is not consistent with the data we see, or why the model doesn't account for this possibility (as the model structure assumes a 100% carriage prevalence it can't formulate this possibility).

3. Following from 2. I am confused about the way the different models / hypotheses are tested. The outcome is the number of infections per strain type across the total population as a function of the parameter governing the mechanism. Is this the best way to test the different models? As far as i understand, for each hypothesis, the size of the parameter is varied between 0 and 50% increase away from the null model (i.e. increasing the virulence from 0 to 50% within the "Virulence" model leads to associated increasing the total numbers of all infections). However, is the dialling up and down of the different parameters what would happen when we observe changing numbers of infections? I don't think so. In reality, there will be a fixed parameter value governing a particular mechanism (e.g. a relative virulence of res vs sens strains with a parameter of 10% will give rise to the number of infections we see). The pertinent question is then - under a mechanism - if we implement interventions that, say, reduce antibiotic use, would any resulting reductions in the number of resistant infections be replaced by a compensatory number of sensitive infections. I think to properly address the issue, Fig 3 would show the number of infections of different types as, e.g., antibiotic use decreases, rather than changing a parameter (e.g. pi) that would in reality be fixed.

4. Finally, following on from 3. I found the results unsurprising given the set up of the model. As far as I understand, the total number of infections is determined exclusively by the virulence rate of each of the demographic groups. None of the models alter the number of people in each demographic compartment, and none alter the virulence rate directly (with the exception of the "Virulence" model). Therefore, there is no mechanism through which any of the models (except for Virulence) could change the total number of infections. I found this result a bit limiting. If the authors had comprehensive conceptual diagrams of how the different models (perhaps as suggested below in the specific comments) I think this main limitation would be clearer.

Some specific suggestions on these points and other comments are given below:

1. The introduction contains many paragraphs of description and justification of the approach. My suggestion would be to give a brief summary of the overall goal of the paper and the method to tackle the goal. The rest can be moved to the Results section by way of a motivation for the Results as the Methods are presented later.

2. 'Open population' - might be better described consistently as the 'community' or 'community population'? (At least for the differential equations it would be nice to see dc/dt rather than do/dt.)

3. I appreciated the description of the models as adjustments away from the neutral 'null' model. I found this a particularly helpful exposition.

4. I found the description of the model particularly confusing as a result of the equations and a repeated but less detailed compartment diagram being in the Supp Mat, with only a text description in the main ms and a simple compartment diagram. My suggestion would be two-fold: first, to add a much more detailed model compartment diagram to the main text; for example, in Colijn et al. (JRSI 2010) the authors use different line types to differentiate between (super)colonisation, clearance, and treatment; further, to create full diagrams that show the different demographic groups and infection compartments, as well as the different model structures (trying to understand Table S3 was a bit of a headache for me). Second, to reformat the equations so they are easier to read. Can anyone in the group perhaps write them in latex? Or at least, not use all the '*' notations and terms like 'sizeo', which makes the terms hard to distinguish and a headache to read.

5. Although parameters are defined in Table S4, there seem to be lots of notation that is not defined. For example, do the equations track the fraction of number of infections? What is 'y'. I couldn't properly assess the model because it was all rather difficult to piece together.

6. Some of the model description I find a bit confusing. For example, I don't know what is meant by the following sentences:

a) "In the SR-compartment, the infection rate is the average of the S- and R-compartment, with infections with resistant or susceptible strains being proportional to the two infection rates."

b) "Without these compartments, introduction of ARB into a population in which everyone could acquire ARB would lead to an equilibrium prevalence of 50%, even in absence of selective advantages." Would this be the single dual infection model (ie X, S, R, SR) as described in Colijn et al. JRSI 2010?

c) "Mechanisms benefitting resistant strains and increasing ARB colonisation prevalence render less people susceptible for acquiring new colonisation with susceptible strains, since we assume a maximum colonisation capacity (RR, SR or SS)"

7. Table S4: Should 'infection rate hospital' just be 'infection rate' ? There are no units given for this - or any of the other parameters.

8. The differential equations don't seem to be complete. For example, I can't find where any equation where the rate of infection, \\pi, which depends on the group (hosp, inpatient or recently hosp), is used. It would be helpful to show compartments called 'infections' (Sens / Res) that show how the number of infections are tracked, especially as this is the main outcome of the paper.

9. "Naturally, introducing costs of resistance without benefits of resistance reduces the incidence of infections caused by ARB and eventually leads to extinction". If you are talking about extinction here, I would suggest rephrasing to carriage (rather than infection), and rename as elimination of resistant strains.

Reviewer #3: Please see the attachment.

**Have the authors made all data and (if applicable) computational code underlying the findings in their manuscript fully available?**

Reviewer #1: Yes

Reviewer #2: Yes

Reviewer #3: Yes

PLOS authors have the option to publish the peer review history of their article (what does this mean?). If published, this will include your full peer review and any attached files.

Reviewer #1: No

Reviewer #2: **Yes: **Katherine Atkins

Reviewer #3: No
---

## [Decision Letter · Decision Letter 1]

15 Dec 2021

Dear Mrs. Godijk,

Thank you very much for submitting your manuscript "Does plasmid-based beta-lactam resistance increase E. coli infections: Modelling addition and replacement mechanisms" for consideration at PLOS Computational Biology. As with all papers reviewed by the journal, your manuscript was reviewed by members of the editorial board and by several independent reviewers. The reviewers appreciated the attention to an important topic. Based on the reviews, we are likely to accept this manuscript for publication, providing that you modify the manuscript according to the review recommendations.

Sincerely,

Roger Dimitri Kouyos

Associate Editor

PLOS Computational Biology

Virginia Pitzer

Deputy Editor-in-Chief

PLOS Computational Biology

[LINK]

Reviewer's Responses to Questions

**Comments to the Authors:**

Reviewer #1: The authors thoroughly addressed the points suggested by the reviewers, nice work. I will just mention one point that I still stumbled upon while reading:

(LL.290-305): In lines 297-299 it is described, that the compartments SS and RR are added to ensure a neutral model, but it is stated that those compartments do not have an obvious meaning. In lines 310-311 it is stated that those compartments are interpreted as high-density compartments. Isn't therefore a biological meaning assigned to the compartments?( The implicit assumptions of within-host state of disturbed and stable flora).I think this paragraph might benefit from a clearer formulation to avoid confusion. Furthermore I think it could be made clearer that the double compartments (SS,SR,RR) are seen as the default colonisation of a "healthy" human, which only changes to S and R due to disturbances. (Maybe starting out with explaining the double compartments and then the single colonised ones? But this is probably just a matter of taste)

(Typo in L 198?: If ARB would not have increaseD?)

Reviewer #2: The authors have responded to my comments very well - thank you for the attention to detail. I especially compliment them on rewriting the models equations and drawing a model diagram. This has made the paper much more user-friendly.

My only remaining comment is really the way that the authors responded to my second point. That is, I was suspicious about the method that tested the hypothesis of replacement of resistant infections. I was arguing that the test of whether resistant strains are in addition or instead of sensitive strains should really be in the context of when the system is perturbed - most notably in the case of introducing an intervention (rather than on a change of parameters that in reality, remain fixed). The authors have included a Table (S9) that assess the impact of these interventions on the number of resistant and sensitive infections with increasing antibiotic control.

Perhaps I am missing something here, but the ‘base case’ scenario - i.e. the first column with ‘full antibiotic use’ indicates different numbers of infections for each mechanism. I don’t believe this is correct way to compare the impact of antibiotic control in the presence of each of the mechanisms. All mechanisms should start with the same base case number of infections and any intervention should therefore be easily compared to each other on the basis of deviation away from this base case.

I would also suggest that this is a figure for ease of understanding - but this is up to the authors’ discretion.

Finally, there is a Typo on line 173.

Reviewer #3: The authors were able to clarify most of the concerns raised before. I still have reservations about how the infection rate being estimated in the manuscript, but I can understand that estimating the infection rate is not an easy task and could make the revision impossible. So I do not have further questions about this manuscript.

**Have the authors made all data and (if applicable) computational code underlying the findings in their manuscript fully available?**

Reviewer #1: Yes

Reviewer #2: Yes

Reviewer #3: Yes

PLOS authors have the option to publish the peer review history of their article (what does this mean?). If published, this will include your full peer review and any attached files.

Reviewer #1: No

Reviewer #2: **Yes: **Katherine Atkins

Reviewer #3: No

Figure Files:

Data Requirements:

Reproducibility:

References:

---

## [Decision Letter · Decision Letter 2]

27 Jan 2022

Dear Mrs. Godijk,

We are pleased to inform you that your manuscript 'Does plasmid-based beta-lactam resistance increase E. coli infections: Modelling addition and replacement mechanisms' has been provisionally accepted for publication in PLOS Computational Biology.

Best regards,

Roger Dimitri Kouyos

Associate Editor

PLOS Computational Biology

Virginia Pitzer

Deputy Editor-in-Chief

PLOS Computational Biology

Reviewer's Responses to Questions

**Comments to the Authors:**

Reviewer #1: The revised version made the model interpretation more accessible for the reader in my opinion. Thanks for clarifying. I have no further comments.

Reviewer #2: I have no more comments. Thank you for the interesting paper.

**Have the authors made all data and (if applicable) computational code underlying the findings in their manuscript fully available?**

Reviewer #1: Yes

Reviewer #2: Yes

PLOS authors have the option to publish the peer review history of their article (what does this mean?). If published, this will include your full peer review and any attached files.

Reviewer #1: No

Reviewer #2: **Yes: **Katherine Atkins

---

## [Editor Report · Acceptance letter]

21 Feb 2022

PCOMPBIOL-D-21-00975R2 

Does plasmid-based beta-lactam resistance increase E. coli infections: Modelling addition and replacement mechanisms

Dear Dr Godijk,

I am pleased to inform you that your manuscript has been formally accepted for publication in PLOS Computational Biology. Your manuscript is now with our production department and you will be notified of the publication date in due course.

With kind regards,

Katalin Szabo
